# Modes of Transport to School and Their Associations with Weight Status: A Cross-Sectional Survey of Students in Shanghai, China

**DOI:** 10.3390/ijerph18094687

**Published:** 2021-04-28

**Authors:** Yuan-Shen Zhu, Zhuo Sun, Dan-Dan Ke, Jia-Qi Yang, Wen-Yun Li, Ze-Qun Deng, Yong-Zhen Li, Min Wu, Li-Ming Wen, Geng-Sheng He

**Affiliations:** 1School of Public Health, Fudan University, Shanghai 200032, China; 18211020127@fudan.edu.cn (Y.-S.Z.); 19211020026@fudan.edu.cn (Z.S.); 19111020026@fudan.edu.cn (J.-Q.Y.); 20111020038@fudan.edu.cn (W.-Y.L.); 17211020137@fudan.edu.cn (Z.-Q.D.); 18211020022@fudan.edu.cn (Y.-Z.L.); wumin@shmu.edu.cn (M.W.); 2Key Lab of Health Technology Assessment, National Health Commission of the People’s Republic of China, Fudan University, Shanghai 200032, China; 3Graduate School of Health and Sports Science, Juntendo University, Chiba 2701695, Japan; sh4218003@juntendo.ac.jp; 4School of Public Health, University of Sydney, Sydney, NSW 2006, Australia; liming.wen@sydney.edu.au

**Keywords:** modes of transport to school, overweight and obesity, population health

## Abstract

*Background:* Over the past two decades, both transport modes as well as overweight/obesity have changed dramatically among students in China, but their relationships are not clear. This study aimed to investigate modes of transport to school and their associations with the weight status of Chinese students. *Methods:* A cross-sectional study was conducted with non-resident students aged 6 to 17 years from all 16 districts across Shanghai, China in October and November 2019. Information about sociodemographic characteristics and the models of travel to school among students was investigated using an online, self-administered, structured questionnaire (or those assisted by their parents). Weight and height were measured by school health workers, and the Chinese standard age adjusted BMI (weight/height^2^) was used to classify students’ weight status. Cumulative logistic regression modelling was used to examine the relationships. *Results:* The main mode of transport to school was an active mode (46.5%, defined as walking, bicycling, or public transport), followed by an inactive mode of transport (30.5%, defined as a car or bicycle as a passenger), and a combination of both modes (23%). About one-third of the students were overweight or obese and 5% were underweight. No statistically significant association between transport modes and weight status was found in this study. *Conclusions:* In Shanghai, close to one-third of children travel to school by an inactive mode of transport. The findings of this study did not support the notion that an active mode to school could be beneficial for preventing overweight/obesity in students in China.

## 1. Introduction

Travel mode choice among students has been widely investigated in developed countries, due to increasing use of cars to commute to school and its health consequence [1,2,3]. With rapid growth in the amount of motor vehicles, the proportion of walking or cycling to school has decreased in China over the past two decades. The China Health and Nutrition Survey indicated that the prevalence of walking or cycling among children and adolescents (aged 6 to 17 years) dropped from 95.8% in 1997 to 69.3% in 2011 [4]. In metropolitan cities of China, such as Shanghai and Beijing, about half (51~56%) of students walked and cycled to or from school, and more than 10% were driven by car to or from school [5,6].

Studies in developed countries have found that commuting to school by car is associated with an obesity status in students [7,8,9]. For example, Wen et al. performed logistic regression analysis on modes of travel to school and a possible association of being overweight and obese among 1362 Australian children aged 10~13 years and reported a positive relationship of being driven to school daily and obesity [7]. Hence, a strategy to prevent overweight/obesity is to encourage children to walk or cycle instead of traveling by car [10,11]. However, the relationship between transport modes and weight status of primary and secondary students is still unclear in China. A multicenter research trial with eight cities (including Shanghai) suggested that active commuting to school (walking or cycling) was correlated with lower risk of obesity [12], while another cross-sectional survey in Shanghai found no significant association [6]. The results of the two studies may be limited by the fact that private cars and public transport were mingled, and also socioeconomic factors of students were not fully considered in their studies. 

To our knowledge, there are few studies exploring the relationship between commuting to school as vehicle passengers and weight status in China. In view of the increasing prevalence of being overweight and obese among adolescents in China, it is particularly important to find out whether the modes of commuting to school can affect the weight status of students [13]. The main aim of this study was to examine modes of transport to school and their associations between modes of transport to school and overweight and obesity among school-age children.

## 2. Materials and Methods

### 2.1. Study Design

We conducted a cross-sectional study in students aged 6 to 18 years from 47 schools across all 16 districts in Shanghai between October and November 2019. The middle school students and parents of primary pupils completed an online self-administered structured questionnaire, which was coded through an anonymous record technique based on schools’ and students’ number in order to meet privacy requirements for conducting surveys in schools.

### 2.2. Study Participants and Data Collection

Participants were recruited by means of a two-stage stratified sampling. In the first stage, a total of 47 schools were selected by convenience sampling. All schools volunteered, which were located across 16 districts of Shanghai, with one primary, one secondary, and one high school in each district (one was a comprehensive school with secondary school and high school). In the second stage, 100 students were randomly chosen from each grade of the participating schools by using stratified cluster sampling. In the 47 participating schools, a total of 19,200 students received an online invitation through their class teachers to take part in the research.

The self-administered structured questionnaire, adapted from the 2010 New South Wales Schools Physical Activity and Nutrition Survey (SPANS 2010) in which most of the key measures have been tested for their reliability and validity, was sent to middle school students via QR code or to parents of primary pupils via WeChat [14]. The questionnaire covered grade, gender, school location, modes of transport to school, and family sociodemographic information. The individual height and body weight data was collected from the Growth and Development Monitoring Program by Shanghai Education Department in September 2019.

### 2.3. Measures

Mode of transport to school was captured by the responses to the question, ‘How do you usually get to school?’ which allowed for multiple responses. There were five alternatives: public transport, cycling, walking, car (including private cars and taxi), and bike as a passenger (including bicycles, motorbikes, and electric bikes).

Body weight and height were measured by the health workers in school, in units of 0.1 cm and 0.1 kg, respectively. Determination of being overweight and obese was based on age adjusted Body Mass Index (BMI, weight/height^2^) with Chinese standard classifications in children and adolescents by Working Group of Obesity in China [15,16]. 

Family sociodemographic information included the educational level and occupation of both parents. A total of three options, ‘up to secondary education (year 12),’ ‘junior college degree,’ and ‘bachelor or above’ were used to determine the family education level. A series of options describing occupation (‘administrative staff,’ ‘technicians,’ ‘clerical staff,’ ‘business people,’ and so on) were listed and participants selected one only.

### 2.4. Data Analysis

The study outcome variable was categorized into four classes: underweight, normal, overweight, and obese. Four modes of transport to school were captured: first, travel by one or multiple type of public transport, bicycle and walking were combined to one mode as “active mode,” second, “car only” was the mode of vehicle passengers, third, “bike as the passenger only” was the mode of bike passengers, and the rest were classified as a mixed mode.

For co-variables, grade was re-grouped into four classes (first to third grade, fourth to fifth one, sixth to ninth one, and tenth to twelfth one), living areas decided by school locations were dichotomized into either a core area or extended area (Appendix A), based on the Shanghai Master Plan (2017–2035) [17], and administrative staff and technicians were considered to be a superior occupation social class, according to the context of occupational prestige in China [17,18]. Relationships between the demographic variables and modes of transport to school were examined using Pearson chi-square tests and the Mantel–Haenszel trend test.

Relationships between the study factor and outcome factor were examined using the bivariate analyses (univariate multinomial logistic regression) and cumulative logistic regression analysis. In the cumulative logistic regression analysis, with regard to the weight status based on BMI (including underweight, normal, overweight, and obese) as the dependent variable, variables were developed for the outcome of interest by using DAGs (Directed Acyclic Graphs) to enter the model compulsorily. Based on the DAGs (Appendix A), we excluded the physical activity variable, as it was a mediator between modes of transport to school and weight status. The variables of screen time, school environment, and diet, as collision, were also excluded. Adjusted odds ratios (AORs) with 95% confidence intervals were then calculated as a measure of the predictive power of transport risk factors for the weight status. All the data processing and statistical analyses were performed in the Stata software v.15.1 (Stata Corp. Texas, TX, USA). A two-sided *p*-value < 0.05 was considered statistically significant.

## 3. Results

### 3.1. Characteristics of the Study Population

Among 19,200 invited students, 15,851 students provided consent to the study with the response rate of 82%. Of the total participants, 4790 were excluded from analysis due to missing height and body weight data. Of the remaining 11,061 potential subjects, 938 were excluded for boarders, of whom 98% were high school students. The remaining 10,123 students formed the sample in the present analysis. 

Among 10,123 participants, the main mode of transport to school was an active mode (46.5%, defined as walking, bicycling, or public transport), followed by being vehicle or bike passengers (30.5%), and by a combination of these modes (23.0%). About 5% of the participants were underweight, 17% were overweight, and 14% were obese. Half of participants (42%) were in the central area of Shanghai with approximately equal proportion in each of the gender and the age groups (Table 1).

### 3.2. The Determinants of Travel Modes

There was a greater proportion of trips to school among girls as the passenger, while the share of active commuting (walking, cycling, or public transport) was higher in boys (Appendix A show the proportion of choice in different areas and grades, which illustrated that car travel instead of active mode was more usual by students living in the surrounding area than the central area as well as being positively associated with grade. Sociodemographic factors showed a significant association with travel modes. Students tended to use cars from the family with a higher parental education level ($\chi_{trend}^{2}$ = 63.47, *p* < 0.001) or superior occupation social class ($\chi_{trend}^{2}$ = 54.19, *p* < 0.001).

### 3.3. The Relationship between Modes of Travel to School and Weight Status

Through univariate analysis, underweight was only significantly associated with gender. Being overweight or obese was significantly associated with gender, age, living area, modes of transport to school, and family educational level (Appendix A).

The results of the cumulative logistic regression are shown in Table 2. No relationship was found between being underweight and modes of transport to school. There was also no significant association with being overweight and obese with regard to modes of travel to school.

## 4. Discussion

The current study comprehensively investigated the modes of transport to school of primary and secondary school students and their associations with students’ weight status in Shanghai, China. Although no statistically significant associations were found between different modes and overall weight status, there was a possible trend that students who travel to school by car were at a lower risk of being overweight or obese. Our finding was consistent with a previous study in Jiangsu Province of China with adult population, in which they found an inverse correlation between active transportation and some health outcomes [18].

In our study, a combination of walking, cycling, and public transport was the most common travel mode across grades and areas in Shanghai, which was consistent with previous studies [5,6,12]. We found that the proportion of travel to school by car (17%) was higher than in other cities in China (average 10% [5]), but still much lower than that in developed countries (51% in AUS [1], 41% in UK [2], 80% in US [3]). It is apparent that travel by car for students living in the surrounding area was more usual compared with the central area (See Appendix A). In certain regions of China, especially metropolises, urban car license control policies have significantly restricted the growth of car ownership and the use of cars on weekdays [19]. Convenient public transportation in the core area also reduced the frequency of car use [19]. Hence, modes of transport to school varied regionally.

The relationship between commuting to school by car and weight status in our study was found to be inconsistent to that of developed countries. Since the childhood obesity increases and the proportion of walking or cycling to school decreases simultaneously, there is a general awareness that commuting to school by car leads to lower levels of physical activity and higher risk of childhood overweight/obesity [7,9]. A cross-sectional survey in German found that increasing walking or cycling distance results in decreasing adolescents’ fat mass [20]. However, the benefit of walking or cycling for weight status is still inconsistent with results by other studies. No association between mode of commuting and being overweight in children aged 10–12 years was found both in a cross-sectional study from eight European countries and a two-year cohort study from America [21,22]. A population-based intervention trial in America also suggested that active commuting (walking, bicycling, or skating) did not provide sufficient amounts of physical activity to attenuate BMI for primary pupils [23]. Studies in China are very scarce. A multicentered National Puberty Research in eight cities (including Shanghai) suggested that active commuting to school (walking or cycling) was correlated with lower risk of adolescents’ obesity [6]. A cross-sectional survey in 11 primary schools of Shanghai did not found any relationships of active commuting and weight status [12]. The grouping of travel modes is always simply dichotomized into “active mode” (walking or cycling) and “passive mode” (public transport, car, or others) in both studies [6,12]. A crude grouping could conceal some environmental factors and weight-related behaviors behind the choice of travel mode.

The students had potentially benefited from the choice of cars, which may be associated with the socio-economic and the policy factors. Under the education policy of “the nearest school”, the home-to-school distance should be within <2 km and commuting time should be, mostly, less than an hour [24]. Travel by cars as the passengers is also chosen for the short-distance trip, which may be led by the link between the parent journey to work and student journey to school. Hence, alteration of the travel mode hardly causes a significant change of the students’ physical activity level under the condition of a short distance from home to school. On the other hand, the choice of car could reflect a relatively high family socio-economic status, to a certain extent [25]. Children from socially and economically advantaged families were less likely to be overweight and obese, which could be partly explained that parents with higher social-economic status could afford extra financial cost of leisure-time physical activities, and, thereby, improve their children’s physical activity levels [26,27]. In addition, students taken by car from school to home may be restricted in their choices and opportunities of sweetmeats or sugary beverages. Possibly, students with their fellows consume snacks and soft drinks on the way back home owing to peer pressure. This needs further investigation. 

Despite no association of modes of travel to school with being overweight and obese in the study, promoting active modes, including walking, cycling, and public transport, is still important given that there are a number of co-benefits associated with active modes of travel, such as reducing air pollution [28,29], traffic noise [30,31], and traffic injury [10,11]. Motor vehicle traffic exposure is a major source of air pollution and traffic noise. Prioritizing walking, cycling, and public transport use would substantially reduce and slow road traffic around the school, and, in turn, protect students from increased exposure risk of road traffic noise and air pollutants [28,30]. Active commuting also makes children have greater opportunities for reflection during the school trip and opportunities for social interactions, and there is a potential to enhance psychological well-being by reducing depressive symptoms [12,32,33]. On the other hand, reducing the dependence of children on cars help them develop the habit of walking and practice their road safety skills, which could lead to fewer traffic injuries [10,11].

The main strength of this study was that our investigation covered all grades in primary and middle schools and all districts in Shanghai with a large sample. The anthropometrics data were measured by health workers instead of self-reports. Nonetheless, the study is subject to some limitations. First, only one private school was included in this survey, which could lead to sample selection bias as students attending private schools were more likely to be of a higher social-economic status than others. Second, social-economic status only consisted of parental educational and occupation without considering family income, which may not reflect the condition precisely. Third, the generalizability of study findings could be limited due to the study only being conducted in Shanghai, China. In addition, the survey did not include information on distance to school or travel time to school, which could also bias the study findings.

## 5. Conclusions

With regard to travel to school, public transport, bicycling and walking are still the main modes of commuting to school, but an increasing number of children (close to one-third) travel to school by an inactive mode of transport in Shanghai, China. The findings of this study did not support the notion that the active mode is beneficial for preventing overweight/obesity in students, and more studies are warranted to confirm such findings.

## Figures and Tables

**Table 1 ijerph-18-04687-t001:** Distribution of study subjects by demographic characteristics.

Characteristics	Total Sample *n* (%)	Central Area *n* (%)	Surrounding Area *n* (%)	*p* Value *
**Gender**				0.107
Male	4995(49.3)	2148(50.3)	2847(48.7)	
Female	5128(50.7)	2124(49.7)	3004(51.3)	
**Age-group**				<0.001
First to third grade	2843(28.1)	1268(29.7)	1575(26.9)	
Fourth to fifth grade	1552(15.3)	610(14.3)	942(16.1)	
Sixth to ninth grade	3331(32.9)	1452(34.0)	1878(32.1)	
Tenth to twelfth grade	2397(23.7)	941(22.0)	1456(24.9)	
**Family educational level**				<0.001
Up to secondary education (year 12)	2328(23.0)	954(22.3)	1374(23.5)	
Junior college degree	2571(25.4)	967(22.6)	1604(27.4)	
Bachelor or above	5224(51.6)	2351(55.1)	2873(49.1)	
**Superior occupation social class**				0.106
Neither	4774(47.2)	2045(47.9)	2729(46.6)	
Only in fathers	2278(22.5)	912(21.3)	1366(23.3)	
Only in mothers	726(7.2)	303(7.1)	423(7.2)	
Both	2345(23.2)	1012(23.7)	1333(22.8)	
**Total**	10,123(100)	4272(42.2)	5851(57.8)	

Note: * Pearson chi-square tests.

**Table 2 ijerph-18-04687-t002:** Association between modes of transport to school and being underweight, overweight, and obese.

Characteristics	Overweight or Obese			Underweight		
	AOR *	95% CI	*p* Value	AOR **	95% CI	*p* Value
**Modes of transport to school**						
Active mode	Reference			Reference		
Vehicle passengers	0.88	0.78–1.00	0.051	1.15	0.89–1.48	0.274
Bike passengers	0.95	0.83–1.08	0.418	1.21	0.91–1.60	0.197
Mixed mode	0.94	0.84–1.05	0.279	0.93	0.73–1.18	0.544
**Gender**						
Male	Reference			Reference		
Female	0.51	0.46–0.55	<0.001	0.77	0.64–0.92	0.004
**Age-group**						
First to third grade	Reference			Reference		
Fourth to fifth grade	1.13	1.00–1.23	0.056	0.86	0.62–1.18	0.337
Sixth to ninth grade	1.00	0.90–1.12	0.960	1.18	0.93–1.51	0.175
Tenth to twelfth grade	0.75	0.66–0.85	<0.001	1.31	1.02–1.70	0.036
**Living area**						
Core	Reference			Reference		
Extended	0.86	0.79–0.94	0.001	0.84	0.70–1.02	0.080
Family educational level						
**Up to secondary education (year 12)**	Reference			Reference		
Junior college degree	0.94	0.83–1.06	0.608	1.01	0.78–1.31	0.927
Bachelor or above	0.84	0.74–0.94	0.003	0.93	0.72–1.19	0.551
**Superior occupation social class**						
Neither	Reference			Reference		
Only in fathers	0.97	0.87–1.09	0.608	1.23	0.98–1.55	0.071
Only in mothers	1.19	1.00–1.41	0.050	1.37	0.97–1.94	0.078
Both	0.94	0.84–1.06	0.329	1.12	0.87–1.43	0.383

Note: * Adjusted Odds ratios (AORs) were adjusted for variables in this table in the cumulative logistic regression analysis. ** Adjusted Odds ratios (AORs) were adjusted for variables in this table in the logistic regression analysis.

## Data Availability

The data presented in this study are available on request from the corresponding author. The data are not publicly available due to ethical.

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
