# Peer review of "Modes of Transport to School and Their Associations with Weight Status: A Cross-Sectional Survey of Students in Shanghai, China"

_ijerph, 2021, doi:10.3390/ijerph18094687_

Round 1

Reviewer 1 Report

The main issue is the lack of description of the instrument used, it must be described more detailed and is it validated?

if not, it is not feasible to apply it to a number of participants like these.

Author Response

Thank you for your comments. Please find attached with my response to your comments.

Regards

Reviewer 2 Report

The topic of this article is the influence of the mode of transport on the weight status among children and adolescents in Shanghai, China. Research on the influence of varies factors on the body weight of children and adolescents is very important. Because both overweight and obesity as well as underweight in childhood are becoming causes of various types of diseases in adulthood.

The article is interesting and written correctly. Unfortunately, the classification of the mode of travel to the school into the active and inactive groups raises doubts. According to the Authors  active mode was walking, bicycling or public transport. It is incomprehensible to qualify public transport as an active form. It is yet, for example, a bus ride. In my opinion, public transport should be considered as a mixed mode of transport to school. He is aware that the Authors will have to correct the statistical calculations, but such a division is methodologically wrong.

Best regards!

Author Response

Thanks for your comments. Please see the attachment.

Regards

Reviewer 3 Report

The authors present results of a cross-sectional study focused on the relationship between how students are transported to school and their weight, or nutritional status (I mean underweight, normal weight, overweight, obesity). Despite the engaging topic, the article acts only as part from a large topic. The topic of article is not anchored. I believe that the authors should focus on the quality of data presentation and documentation (or precisious describe) of contexts. In terms of content, I perceive the manuscript as poor quality.
1) I consider the abstract to be very inappropriately formulated. After reading it, I am asking why the authors publish such research. I think that the final passages of the results and especially the conclusion of the abstract significantly reduce the contribution of work and devalue their own work.
2) The introduction provides the necessary data to outline the issue.
3) The methodology is strong part of this manuscript. It is clearly described and structured. The authors justify the evaluation of nutritional status by BMI, however, I believe that it is more appropriate to choose percentile growth charts in the pediatric population. 
4) I consider the presentation of data and the method of their processing to be unhappily chosen. The authors have at their disposal very interesting data coming from a large number of respondents, which can be well presented with a suitable grasp. However, other variables are not taken into account in data presentation and context search. The mode of transport is undoubtedly also influenced by the age of the child (student). I do not understand why in the results the authors associate walking, bicycling or public transport under the active mode of transport to school. This is not logical from my point of view. Public transport has other characteristics, I am not entirely sure whether it can be perceived as an active way if a student travels by public transport (ie by train, bus, etc.). It can be confusing.
Overall, I consider the presentation of data to be insufficient, very austere and confusing for readers.
5) The discussion contains a lot of interesting information. The authors prove that they address the issue. But they do not work much with the results of their work. The discussion seems more like an introduction in some parts. It brings insight into the issue.
6) The conclusion is insufficient. It is not substantiated by the results of research, it does not imply an incentive, an intention for the future.

Author Response

Thanks for your comments. Please find attached with my response to your comments.

Regards

Round 2

Reviewer 1 Report

the authors have adressed the main issues to be changed.